# Analysis of three-dimensional unsaturated-saturated flow induced by localized recharge in unconfined aquifers

Chia-Hao Chang[1], Ching-Sheng Huang[2], Hund-Der Yeh[1]

[1]Institute of Environmental Engineering, National Chiao Tung University, Hsinchu, Taiwan

[2]State Key Laboratory of Hydrology-Water Resources and Hydraulic Engineering, Center for Global Change and Water Cycle, Hohai University, Nanjing, China

*Correspondence to*: Hund-Der Yeh (hdyeh@mail.nctu.edu.tw)

**Abstract.** In the process of groundwater recharge, surface water usually enters an aquifer by passing an overlying unsaturated zone. Little attention has been given to the development of analytical solutions to a coupled unsaturated-saturated flow model

due to localized recharge up to now. This paper develops a mathematical model to depict three-dimensional transient unsaturated-saturated flow in an unconfined aquifer with localized recharge on the ground surface. The model contains Richards' equation for unsaturated flow, a flow equation for saturated formation, and the Gardner constitutive model describing the behavior of unsaturated soil properties. Both flow equations are coupled through the continuity conditions of the head and flux at the water table. The semi-analytical solution to the coupled flow model is derived by the methods of Laplace transform

and Fourier cosine transform. A sensitivity analysis is performed to explore the head response to the change in each of the aquifer parameters. A quantitative tool is presented to assess the recharge efficiency signifying the percentage of the water from the recharge to the aquifer. We found that the effect of unsaturated flow on the saturated hydraulic head is negligible if two criteria associated with the unsaturated soil properties and initial aquifer thickness are satisfied. The head distributions predicted from the present solution match well with those from finite difference simulations. The predictions of the present

solution also agree well with the observed data from a field experiment at an artificial recharge pond in Fresno County, California.

## 1 Introduction

Understanding the effect of water flow due to recharge from a surface water body such as precipitation, lake, or artificial pond on the groundwater flow system is important in water resource planning and management (e.g., Wang et al., 2010; Siltecho et

al., 2015; Yang et al., 2015; Scudeler et al., 2016; Shi et al., 2016). The subsurface soil formation may be divided to unsaturated and saturated zones depending on the water saturation in void spaces of the soils. In the recharge process, the surface water may infiltrate and flow through the unsaturated zone and then arrives at the water table of the saturated zone (i.e., aquifers). Chang et al. (2016) reviewed analytical solutions describing the spatiotemporal distributions of groundwater mounds caused by localized recharge on the ground surface. They classified 17 solutions in a tabular form with flow dimensions as well as six

headings of references, aquifer domain, aquifer boundary conditions, recharge region, recharge rate, and remarks. However, those solutions they reviewed all neglect the process of infiltration in the unsaturated zone and assume that the surface water directly recharges the saturated zone.

Solving Richards' equation (Richards, 1931) analytically for unsaturated flow is tricky owing to its nonlinearity. Gardner (1958) presented a model to express the relative hydraulic conductivity as an exponential function of the pressure head in unsaturated soils. Analytical methods for developing the solution to Richards' equation mostly rely on the use of linearization based on Gardner's model. Many articles used such an approach to study flow in an unsaturated zone with infiltration from a variety of surface water bodies (see, e.g., Huang and Wu, 2012; Wu et al., 2013; Wang and Li, 2015). Those articles neglected the presence of underlying aquifer and treated its water table as the lower boundary with a condition of constant pressure head (e.g., Huang and Wu, 2012) or water content (e.g., Chen et al., 2001b). For 1D downward flow, Srivastava and Yeh (1991) discussed distributions of the pressure head and water content in two distinct unsaturated soil layers with a constant surface flux. Chen et al. (2001a) examined the water content in an unsaturated medium with an arbitrary time-varying surface flux, by extending Warrick's (1975) solution for a flux consisting of step functions of time. Later, Wu et al. (2012) did a similar work to Srivastava and Yeh (1991) but additionally considered the deformation of the two-layer soils caused by the change of the pore-water pressure in the soils due to the surface flux. For 2D flow in a vertical plane, Batu (1980) analyzed steady-state flow net affected by an array of strip surface sources with two different infiltration rates. Protopapas and Bras (1991) focused on the transient pressure head due to a uniform strip source with a finite width and infinite length on the ground surface. For 3D flow, Chen et al. (2001b) investigated the water content induced by a surface source with an arbitrary spatiotemporal infiltration rate. Tracy (2007) studied the pressure head distribution in a cuboid soil sample with localized recharge over a rectangular area on the top. The sides of the sample are under either the Dirichlet or no-flow boundary condition.

Abovementioned solutions are applicable to either the case of saturated flow in aquifers recharged directly by surface water or the case of unsaturated flow due to surface water infiltration. So far little has been known about the combination of saturated and unsaturated flows that represents a typical process of recharge to the aquifer. This paper aims at developing a mathematical model for describing 3D transient unsaturated-saturated flow in an unconfined aquifer with localized recharge. Richards' equation along with Gardner's model is adopted to delineate unsaturated flow between the ground surface and the water table. The 3D groundwater flow equation is employed to depict saturated flow in the aquifer. Richards' equation is coupled with the saturated flow equation via the continuity conditions of the head and flux at the water table. Such a coupled flow model has been proposed to investigate pumping drawdown problems by several articles (e.g., Mathias and Bulter, 2006; Tartakovsky and Neuman, 2007; Mishra and Neuman, 2010; Mishra and Neuman, 2011). They treated an extraction well as a line sink in the aquifer while we consider the localized recharge as a plane source to the aquifer. The coupled flow model in their studies is 2D written in cylindrical coordinates while that in ours is 3D expressed in Cartesian coordinates. In addition, their solutions are obtained by the Hankel transform, but ours is based on the Fourier cosine transform. The present work aims to investigate the spatiotemporal distribution of the hydraulic head due to localized recharge from the ground surface. The semi-analytical solution for the hydraulic head is obtained by the Laplace transform and the Fourier cosine transform. A finite difference

solution is built to check the correctness of the present solution. The effect of the unsaturated zone on the head in the saturated aquifer is explored by the present solution. The water quantity from the localized recharge to the aquifer is analyzed. The sensitivity analysis is executed to examine the head response to the variation in each of the aquifer parameters. Application of the present solution to a field experiment of artificial recharge is also provided.

## 5   2 Methodology

### 2.1 Mathematical model

Consider an unconfined aquifer system with localized recharge over a rectangular area on the ground surface of the system. The origin of the Cartesian coordinate system locates at the center of the recharge area as illustrated in Fig. 1a. The area has a size of $2l$ by $2w$ on $x$-$y$ plane. The shortest distance between an observation point $(x, y)$ and a point $(x_e, y_e)$ on the edge of the

area is defined as $d = \min(\sqrt{(x - x_e)^2 + (y - y_e)^2})$. The initial water table separates the unsaturated and saturated zones as shown in Fig. 1b and is chosen as the reference datum of the coordinate system. The initial thicknesses of the unsaturated and saturated zones prior to the recharge are denoted as $b$ and $B$, respectively.

The mathematical model for the aquifer system comprises two simultaneous equations for unsaturated and saturated flows. The equation for saturated flow in homogeneous and anisotropic aquifers is expressed as

$$K_x \frac{\partial^2 h}{\partial x^2} + K_y \frac{\partial^2 h}{\partial y^2} + K_z \frac{\partial^2 h}{\partial z^2} = S_s \frac{\partial h}{\partial t} \text{ for } -B \leq z \leq 0 \tag{1}$$

where $h(x, y, z, t)$ is the hydraulic head in the saturated zone; $t$ is elapsed time since recharge began; $K_x$, $K_y$, and $K_z$ are respectively the saturated hydraulic conductivities in $x$-, $y$-, and $z$-directions; $S_s$ is the specific storage. Richards' equation for unsaturated flow is expressed as (Richards, 1931)

$$K_x \frac{\partial}{\partial x}\left[k_r(\phi)\frac{\partial \phi}{\partial x}\right] + K_y \frac{\partial}{\partial y}\left[k_r(\phi)\frac{\partial \phi}{\partial y}\right] + K_z \frac{\partial}{\partial z}\left[k_r(\phi)\frac{\partial \phi}{\partial z}\right] = C(\phi)\frac{\partial \phi}{\partial t} \tag{2}$$

where $\phi(x, y, z, t)$ is the hydraulic head in the unsaturated zone. The relative hydraulic conductivity $k_r(\phi)$ and specific moisture capacity $C(\phi)$ are defined by the Gardner constitutive model (Gardner, 1958) as

$$k_r(\phi) = e^{a(\phi - z)} \tag{3}$$

and

$$C(\phi) = aS_y e^{a(\phi - z)} \tag{4}$$

where $S_y$ is the specific yield and $a$ is the unsaturated exponent related to the pore-size distribution of a medium ranging from 0.2 to 5 m$^{-1}$ (Philip, 1969). Substituting Eqs. (3) and (4) into Eq. (2) leads to

$$K_x \frac{\partial^2 \phi}{\partial x^2} + K_y \frac{\partial^2 \phi}{\partial y^2} + K_z \frac{\partial^2 \phi}{\partial z^2} - aK_z \frac{\partial \phi}{\partial z} + a \left[ K_x \left( \frac{\partial \phi}{\partial x} \right)^2 + K_y \left( \frac{\partial \phi}{\partial y} \right)^2 + K_z \left( \frac{\partial \phi}{\partial z} \right)^2 \right] = aS_y \frac{\partial \phi}{\partial t} \tag{5}$$

It is essentially nonlinear and difficultly solved by analytical methods. Kroszynski and Dagan (1975) employed the approach of perturbation expansion to simplify Richards' equation as a first-order linearized equation and developed an approximate solution for unsaturated-saturated flow induced by well pumping. The approach is extensively used in many studies on unsaturated-saturated flow (e.g., Mathias and Butler, 2006; Tartakovsky and Neuman, 2007; Mishra et al., 2012; Liang et al., 2017a). The linearized version of Richards' equation is written as

$$K_x \frac{\partial^2 \phi}{\partial x^2} + K_y \frac{\partial^2 \phi}{\partial y^2} + K_z \frac{\partial^2 \phi}{\partial z^2} - aK_z \frac{\partial \phi}{\partial z} = aS_y \frac{\partial \phi}{\partial t} \text{ for } 0 \leq z \leq b \tag{6}$$

The initial conditions for those two zones are

$$\phi = h = 0 \text{ at } t = 0. \tag{7}$$

Because of symmetry of the recharge area along the $x$ and $y$ axes, the first quadrant (i.e., $x \geq 0$ and $y \geq 0$) of the flow domain is considered. Thus, all the horizontal outer boundaries are specified as the no-flow condition expressed as

$$\frac{\partial \phi}{\partial \mathbf{x}} = \frac{\partial h}{\partial \mathbf{x}} = 0 \text{ at } \mathbf{x} = 0 \tag{8}$$

$$\lim_{\mathbf{x} \to \infty} \frac{\partial \phi}{\partial \mathbf{x}} = \lim_{\mathbf{x} \to \infty} \frac{\partial h}{\partial \mathbf{x}} = 0 \tag{9}$$

where $\mathbf{x} \in (x, y)$. The top boundary condition for the recharge area is denoted as

$$K_z e^{-az} \frac{\partial \phi}{\partial z} = I[\mathrm{H}(x) - \mathrm{H}(x - l)][\mathrm{H}(y) - \mathrm{H}(y - w)] \text{ at } z = b \tag{10}$$

where $I$ is a constant recharge rate and H( ) is the Heaviside step function. Note that Eq. (10) can be written as $K_z \exp(-az) \partial \phi / \partial z = I$ inside the recharge area $0 \leq x \leq l$ and $0 \leq y \leq w$ and denoted as $\partial \phi / \partial z = 0$ outside that area. The impermeable boundary condition at the aquifer bottom is written as

$$\frac{\partial h}{\partial z} = 0 \text{ at } z = -B. \tag{11}$$

The two continuity requirements of the hydraulic head and flux at the water table are expressed, respectively, as

$$\phi = h \text{ at } z = 0 \tag{12}$$

and

$$\frac{\partial \phi}{\partial z} = \frac{\partial h}{\partial z} \text{ at } z = 0. \tag{13}$$

The continuity conditions are valid when the water table change is less than 50 % of the initial saturated aquifer thickness, which is certified by a Hele-Shaw experiment (Marino, 1967).

Define the dimensionless variables and parameters as follows

$$\bar{h} = \frac{h}{B}, \bar{\phi} = \frac{\phi}{B}, \bar{x} = \frac{x}{d}, \bar{y} = \frac{y}{d}, \bar{z} = \frac{z}{B}, \bar{t} = \frac{K_x t}{S_s d^2}, \bar{l} = \frac{l}{d}, \bar{w} = \frac{w}{d}, \bar{b} = \frac{b}{B}, \kappa_z = \frac{K_z d^2}{K_x B^2}, \kappa_y = \frac{K_y}{K_x}, \alpha = aB, \sigma = \frac{S_y}{S_s B}, \xi = \frac{I}{K_z} \quad (14)$$

where the overbar represents a dimensionless variable or parameter. According to Eq. (14), the unsaturated-saturated flow model is rewritten as

$$\frac{\partial^2 \bar{\phi}}{\partial \bar{x}^2} + \kappa_y \frac{\partial^2 \bar{\phi}}{\partial \bar{y}^2} + \kappa_z \frac{\partial^2 \bar{\phi}}{\partial \bar{z}^2} - \alpha \kappa_z \frac{\partial \bar{\phi}}{\partial \bar{z}} = \alpha \sigma \frac{\partial \bar{\phi}}{\partial \bar{t}} \text{ for } 0 \leq \bar{z} \leq \bar{b} \tag{15}$$

$$\frac{\partial^2 \bar{h}}{\partial \bar{x}^2} + \kappa_y \frac{\partial^2 \bar{h}}{\partial \bar{y}^2} + \kappa_z \frac{\partial^2 \bar{h}}{\partial \bar{z}^2} = \frac{\partial \bar{h}}{\partial \bar{t}} \text{ for } -1 \leq \bar{z} \leq 0 \tag{16}$$

$$\bar{\phi} = \bar{h} = 0 \text{ at } \bar{t} = 0 \tag{17}$$

$$\frac{\partial \bar{\phi}}{\partial \bar{x}} = \frac{\partial \bar{h}}{\partial \bar{x}} = 0 \text{ at } \bar{x} = 0 \tag{18}$$

$$\lim_{\bar{x} \to \infty} \frac{\partial \bar{\phi}}{\partial \bar{x}} = \lim_{\bar{x} \to \infty} \frac{\partial \bar{h}}{\partial \bar{x}} = 0 \tag{19}$$

$$\exp(-\alpha \bar{z}) \frac{\partial \bar{\phi}}{\partial \bar{z}} = \xi [H(\bar{x}) - H(\bar{x} - \bar{l})][H(\bar{y}) - H(\bar{y} - \bar{w})] \text{ at } \bar{z} = \bar{b} \tag{20}$$

$$\frac{\partial \bar{h}}{\partial \bar{z}} = 0 \text{ at } \bar{z} = -1 \tag{21}$$

$$\bar{\phi} = \bar{h} \text{ at } \bar{z} = 0 \tag{22}$$

$$\frac{\partial \bar{\phi}}{\partial \bar{z}} = \frac{\partial \bar{h}}{\partial \bar{z}} \text{ at } \bar{z} = 0 \tag{23}$$

where $\bar{\mathbf{x}} \in (\bar{x}, \bar{y})$.

## 2.2 Laplace domain solution

The unsaturated-saturated flow model composed of Eqs. (15) − (23) is solved by the methods of Laplace and Fourier cosine transforms. The Laplace transform is defined as

$$\tilde{f} = \int_0^\infty \bar{f} \exp(-p\bar{t}) \mathrm{d}\bar{t} \tag{24}$$

with the property that

$$\int_0^\infty \frac{\partial \bar{f}}{\partial \bar{t}} \exp(-p\bar{t}) \mathrm{d}\bar{t} = p\tilde{f} - \bar{f}|_{\bar{t}=0} \tag{25}$$

where $\tilde{f} \in (\tilde{\phi}, \tilde{h})$ represents the dimensionless hydraulic head in the Laplace domain, $p$ is the Laplace transform parameter, $\bar{f} \in (\bar{\phi}, \bar{h})$, and $\bar{f}|_{\bar{t}=0} = 0$ is from Eq. (17). Using Eqs. (24) and (25) converts $\bar{\phi}(\bar{x}, \bar{y}, \bar{z}, \bar{t})$ into $\tilde{\phi}(\bar{x}, \bar{y}, \bar{z}, p)$, $\partial\bar{\phi}/\partial\bar{t}$ into $p\tilde{\phi}$, $\bar{h}(\bar{x}, \bar{y}, \bar{z}, \bar{t})$ into $\tilde{h}(\bar{x}, \bar{y}, \bar{z}, p)$, $\partial\bar{h}/\partial\bar{t}$ into $p\tilde{h}$, and $\xi$ into $\xi/p$. The model then becomes

$$\frac{\partial^2 \tilde{\phi}}{\partial \bar{x}^2} + \kappa_y \frac{\partial^2 \tilde{\phi}}{\partial \bar{y}^2} + \kappa_z \frac{\partial^2 \tilde{\phi}}{\partial \bar{z}^2} - \alpha\kappa_z \frac{\partial \tilde{\phi}}{\partial \bar{z}} = \alpha\sigma p\tilde{\phi} \tag{26}$$

and

$$\frac{\partial^2 \tilde{h}}{\partial \bar{x}^2} + \kappa_y \frac{\partial^2 \tilde{h}}{\partial \bar{y}^2} + \kappa_z \frac{\partial^2 \tilde{h}}{\partial \bar{z}^2} = p\tilde{h} \tag{27}$$

subject to the transformed boundary conditions of $\lim\limits_{\bar{\mathbf{x}}\to 0,\infty} \partial\tilde{\phi}/\partial\bar{\mathbf{x}} = \lim\limits_{\bar{\mathbf{x}}\to 0,\infty} \partial\tilde{h}/\partial\bar{\mathbf{x}} = 0$ with $\bar{\mathbf{x}} \in (\bar{x}, \bar{y})$, $\exp(-\alpha\bar{z}) \partial\tilde{\phi}/\partial\bar{z} = (\xi/p)[\mathrm{H}(\bar{x}) - \mathrm{H}(\bar{x} - \bar{l})][\mathrm{H}(\bar{y}) - \mathrm{H}(\bar{y} - \bar{w})]$ at $\bar{z} = \bar{b}$, and $\partial\tilde{h}/\partial\bar{z} = 0$ at $\bar{z} = -1$. Moreover, the transformed continuity conditions are $\tilde{\phi} = \tilde{h}$ and $\partial\tilde{\phi}/\partial\bar{z} = \partial\tilde{h}/\partial\bar{z}$ at $\bar{z} = 0$.

Afterward, one may take the double Fourier cosine transform that provides

$$\hat{f} = \int_0^\infty \int_0^\infty \tilde{f} \cos(\omega_1 \bar{x}) \cos(\omega_2 \bar{y}) \, \mathrm{d}\bar{x}\mathrm{d}\bar{y} \tag{28}$$

and

$$\int_0^\infty \int_0^\infty \left(\frac{\partial^2 \tilde{f}}{\partial \bar{x}^2} + \kappa_y \frac{\partial^2 \tilde{f}}{\partial \bar{y}^2}\right) \cos(\omega_1 \bar{x}) \cos(\omega_2 \bar{y}) \, \mathrm{d}\bar{x}\mathrm{d}\bar{y} = -(\omega_1^2 + \kappa_y\omega_2^2)\hat{f} \tag{29}$$

where $\hat{f} \in (\hat{\phi}, \hat{h})$ represents the dimensionless hydraulic head in the Fourier domain; $\omega_1$ and $\omega_2$ are the Fourier cosine

transform parameters. The transform converts $\tilde{\phi}(\bar{x}, \bar{y}, \bar{z}, p)$ into $\hat{\phi}(\omega_1, \omega_2, \bar{z}, p)$, $\tilde{h}(\bar{x}, \bar{y}, \bar{z}, p)$ into $\hat{h}(\omega_1, \omega_2, \bar{z}, p)$, $\partial^2\tilde{\phi}/\partial\bar{x}^2 + \kappa_y(\partial^2\tilde{\phi}/\partial\bar{y}^2)$ into $-(\omega_1^2 + \kappa_y\omega_2^2)\hat{\phi}$, $\partial^2\tilde{h}/\partial\bar{x}^2 + \kappa_y(\partial^2\tilde{h}/\partial\bar{y}^2)$ into $-(\omega_1^2 + \kappa_y\omega_2^2)\hat{h}$, and $(\xi/p)[\mathrm{H}(\bar{x}) - \mathrm{H}(\bar{x} - \bar{l})][\mathrm{H}(\bar{y}) - \mathrm{H}(\bar{y} - \bar{w})]$ into $\xi \sin(\omega_1\bar{l}) \sin(\omega_2\bar{w})/(p\omega_1\omega_2)$. Equations (26) and (27) hence become ordinary differential equations in terms of $\bar{z}$ denoted, respectively, as

$$\kappa_z \frac{\partial^2 \hat{\phi}}{\partial \bar{z}^2} - \alpha\kappa_z \frac{\partial \hat{\phi}}{\partial \bar{z}} - (\alpha\sigma p + \omega_1^2 + \kappa_y\omega_2^2)\hat{\phi} = 0 \tag{30}$$

and

$$\kappa_z \frac{\partial^2 \hat{h}}{\partial \bar{z}^2} - (p + \omega_1^2 + \kappa_y\omega_2^2)\hat{h} = 0. \tag{31}$$

Similarly, the transformed boundary conditions are expressed as

$$\frac{\partial \hat{h}}{\partial \bar{z}} = 0 \text{ at } \bar{z} = -1 \tag{32}$$

and

$$\exp(-\alpha \bar{z}) \frac{\partial \hat{\phi}}{\partial \bar{z}} = \xi \sin(\omega_1 \bar{l}) \sin(\omega_2 \bar{w}) / (p \omega_1 \omega_2) \text{ at } \bar{z} = \bar{b}. \tag{33}$$

The transformed continuity conditions are written as

$$\hat{\phi} = \hat{h} \text{ at } \bar{z} = 0 \tag{34}$$

and

$$\frac{\partial \hat{\phi}}{\partial \bar{z}} = \frac{\partial \hat{h}}{\partial \bar{z}} \text{ at } \bar{z} = 0. \tag{35}$$

Solving Eqs. (30) and (31) subject to Eqs. (32) − (35) and then taking the inverse Fourier cosine transform leads to the solutions in the Laplace domain written as

$$\tilde{\phi}(\bar{x}, \bar{y}, \bar{z}, p) = \frac{4}{\pi^2} \int_0^\infty \int_0^\infty R_e(\bar{x}, \bar{y}, \omega_1, \omega_2) \Omega_\phi(\omega_1, \omega_2, \bar{z}, p) \, d\omega_1 d\omega_2 \text{ for } 0 \le \bar{z} \le \bar{b} \tag{36a}$$

and

$$\tilde{h}(\bar{x}, \bar{y}, \bar{z}, p) = \frac{4}{\pi^2} \int_0^\infty \int_0^\infty R_e(\bar{x}, \bar{y}, \omega_1, \omega_2) \Omega_h(\omega_1, \omega_2, \bar{z}, p) \, d\omega_1 d\omega_2 \text{ for } -1 \le \bar{z} \le 0 \tag{36b}$$

with

$$\Omega_\phi = \frac{4}{p(\mu_1 + \mu_2)} \exp\left[\frac{\alpha \bar{z} + (\alpha + \lambda_2)\bar{b}}{2}\right] [\lambda_2 \cosh \lambda_1 \cosh\left(\frac{\lambda_2 \bar{z}}{2}\right) + (2\lambda_1 \sinh \lambda_1 - \alpha \cosh \lambda_1) \sinh\left(\frac{\lambda_2 \bar{z}}{2}\right)] \tag{36c}$$

$$\Omega_h = \frac{4}{p(\mu_1 + \mu_2)} \lambda_2 \exp\left[\frac{(\alpha + \lambda_2)\bar{b}}{2}\right] \cosh[(1 + \bar{z})\lambda_1] \tag{36d}$$

$$R_e = \begin{cases} \xi \sin(\omega_1 \bar{l}) \sin(\omega_2 \bar{w}) \cos(\omega_1 \bar{x}) \cos(\omega_2 \bar{y}) / (\omega_1 \omega_2) \text{ for } \omega_1 \ne 0 \text{ and } \omega_2 \ne 0 \\ \xi \bar{w} \sin(\omega_1 \bar{l}) \cos(\omega_1 \bar{x}) / \omega_1 \text{ for } \omega_1 \ne 0 \text{ and } \omega_2 = 0 \\ \xi \bar{l} \sin(\omega_2 \bar{w}) \cos(\omega_2 \bar{y}) / \omega_2 \text{ for } \omega_1 = 0 \text{ and } \omega_2 \ne 0 \\ \xi \bar{w} \bar{l} \text{ for } \omega_1 = 0 \text{ and } \omega_2 = 0 \end{cases} \tag{36e}$$

$$\mu_1 = [\exp(\bar{b}\lambda_2) - 1](\lambda_2^2 - \alpha^2) \cosh \lambda_1 \tag{36f}$$

$$\mu_2 = 2\lambda_1[(\lambda_2 + \alpha) \exp(\bar{b}\lambda_2) + \lambda_2 - \alpha] \sinh \lambda_1 \tag{36g}$$

$$\lambda_1 = \sqrt{(p + \omega_1^2 + \kappa_y \omega_2^2)/\kappa_z}; \quad \lambda_2 = \sqrt{\alpha^2 + 4(\alpha \sigma p + \omega_1^2 + \kappa_y \omega_2^2)/\kappa_z}. \tag{36h}$$

Notice that Eq. (36a) is the solution for unsaturated flow while Eq. (36b) is that for saturated flow. The inverse Laplace transform to both solutions may not be tractable. The numerical inversion of Laplace transform proposed by Stehfest (1970)
20  is therefore used to obtain time-domain results of the solutions. The double integrals in the solutions can be evaluated numerically by the Gaussian quadrature (e.g., Gerald and Wheatley, 2004) using the Matlab built-in function dblquad (Gilat and Subramaniam, 2007) or the Mathematica built-in function NIntegrate (Wolfram, 1996).

## 2.3 Solution for transient recharge rate

The present solution can be applied to the problem of time-varying recharge rates based on Duhamel's integral (Bear, 1979, p.158). The dimensionless transient head solution $\bar{g}_t$ subject to the dimensionless time-varying recharge rate $\xi_t(\bar{t})$ can be expressed as

$$\bar{g}_t = \bar{g}_0 + \int_0^{\bar{t}} \frac{\partial \xi_t(\tau)}{\partial \tau} \bar{g}(\bar{t} - \tau)\, d\tau \tag{37}$$

where $\tau$ is a dummy variable, $\bar{g}_0$ denotes $\bar{\phi}$ or $\bar{h}$ for the initial dimensionless recharge rate $\xi_t(\bar{t} = 0)$, and $\bar{g}(\bar{t} - \tau)$ represents $\bar{\phi}$ or $\bar{h}$ with $\bar{t}$ replaced by $\bar{t} - \tau$. If Eq. (37) is not an integrable function, it can be evaluated numerically through the discretization method that (Singh, 2005)

$$\bar{g}_N = \bar{g}_0 + \sum_{i=1}^{N} \frac{\Delta \xi_i}{\Delta \bar{t}} G(N - i + 1) \tag{38a}$$

$$\Delta \xi_i = \xi_i - \xi_{i-1} \tag{38b}$$

$$G(M) = \int_0^{\Delta \bar{t}} \bar{g}(M\Delta \bar{t} - \tau)\,d\tau \tag{38c}$$

where $\bar{g}_N$ signifies the dimensionless head solution at $\bar{t} = \Delta \bar{t} \times N$; $\Delta \bar{t}$ is a dimensionless time step; $G(M)$ is called ramp kernel; $\xi_i$ and $\xi_{i-1}$ are respectively dimensionless recharge rates at $\bar{t} = \Delta \bar{t} \times i$ and $\bar{t} = \Delta \bar{t} \times (i - 1)$.

## 2.4 Recharge efficiency

The percentage of the water from the localized recharge reaching the water table is defined as recharge efficiency ($RE$) (Munevar and Marino, 1999) written as

$$RE(t) = K_z \int_0^{\infty} \int_0^{\infty} \frac{\partial h}{\partial z}\, dx\, dy / (I\, l\, w) \text{ at } z = 0 \tag{39}$$

where the denominator $I \times l \times w$ is the volumetric rate of the water entering the aquifer system from the recharge, and the double integral is the sum of the infiltration flux at the water table. According to the dimensionless quantities defined in Eq. (14), Eq. (39) becomes

$$\widetilde{RE}(p) = \int_0^{\infty} \int_0^{\infty} \frac{\partial \tilde{h}}{\partial \bar{z}}\, d\bar{x}\, d\bar{y} / (\xi\, \bar{l}\, \bar{w}) \text{ at } \bar{z} = 0 \tag{40}$$

where $\widetilde{RE}$ represents $RE$ in the Laplace domain and $\tilde{h}$ is defined in Eq. (36b). The $RE$ increases from zero to a value equal to or below unity. The infiltration process does not affect the water table when $RE = 0$. On the other hand, the water from the surface recharge totally arrives at the aquifer when $RE = 1$.

## 2.5 Sensitivity analysis

The sensitivity analysis is commonly used to assess the change of the hydraulic head in response to a small change in a hydraulic parameter. The normalized sensitivity coefficient based on the present solution is defined as

$$S_{i,t} = \frac{\partial O}{\partial P_i / P_i} \tag{41}$$

where $O$ represents the present solution for the unsaturated or saturated flow and $P_i$ is the $i$-th parameter. Equation (41) can be approximated as

$$S_{i,t} = \frac{O(P_i + \Delta P_i) - O(P_i)}{\Delta P_i / P_i} \tag{42}$$

where $\Delta P_i$ is an increment set to $10^{-3} P_i$ (Yeh et al., 2008). Note that a large value of $|S_{i,t}|$ indicates that the head is sensitive to the change in the target parameter.

## 2.6 Finite difference solution

An iterative algorithm based on an implicit finite difference approximation to Eq. (5) is developed to solve the nonlinear unsaturated-saturated flow model. Figure 2 shows the finite difference grids in the simulation domain of $0 \leq x \leq 500$ m, $0 \leq y \leq 500$ m, and -20 m $\leq z \leq 10$ m discretized by a non-uniform grid with small grid sizes near the recharge area of $0 \leq x \leq 50$ m and $0 \leq y \leq 50$ m and large grid sizes away from that area. The domain falls in the first quadrant due to symmetrical flow to $x$-axis and $y$-axis. The saturated thickness is 20 m and the unsaturated thickness is 10 m. All the boundaries except the recharge region are therefore under the no-flow condition. Equation (5) is approximated as

$$K_x \left( \frac{\Delta x_w}{\Delta X} \phi_{i+1,j,k}^{m+1} - \frac{2}{\Delta x_w \Delta x_e} \phi_{i,j,k}^{m+1} + \frac{\Delta x_e}{\Delta X} \phi_{i-1,j,k}^{m+1} \right) + K_y \left( \frac{\Delta y_s}{\Delta Y} \phi_{i,j+1,k}^{m+1} - \frac{2}{\Delta y_s \Delta y_n} \phi_{i,j,k}^{m+1} + \frac{\Delta y_n}{\Delta Y} \phi_{i,j-1,k}^{m+1} \right) + \frac{K_z}{\Delta z^2} \left( \phi_{i,j,k+1}^{m+1} - 2\phi_{i,j,k}^{m+1} + \right.$$

$$\left. \phi_{i,j,k-1}^{m+1} \right) - \frac{a K_z}{\Delta z} \left( \phi_{i,j,k+1}^{m+1} - \phi_{i,j,k}^{m+1} \right) + a \left[ K_x \left( \Delta x_w \phi_{i+1,j,k}^{m+1} - (\Delta x_w + \Delta x_e)\phi_{i,j,k}^{m+1} + \Delta x_e \phi_{i-1,j,k}^{m+1} \right)^2 + K_y \left( \Delta y_s \phi_{i,j+1,k}^{m+1} - (\Delta y_s + \right.$$

$$\left. \Delta y_n)\phi_{i,j,k}^{m+1} + \Delta y_n \phi_{i,j-1,k}^{m+1} \right)^2 + \frac{K_z}{\Delta z^2} \left( \phi_{i,j,k+1}^{m+1} - \phi_{i,j,k}^{m+1} \right)^2 \right] = \frac{a S_y}{\Delta t} \left( \phi_{i,j,k}^{m+1} - \phi_{i,j,k}^{m} \right) \tag{43a}$$

$$\Delta X = \Delta x_w \Delta x_e (\Delta x_w + \Delta x_e)/2 \tag{43b}$$

$$\Delta Y = \Delta y_s \Delta y_n (\Delta y_s + \Delta y_n)/2 \tag{43c}$$

where $\phi_{i,j,k}^{m}$ is the hydraulic head in the unsaturated zone at a nodal point $(i, j, k)$; superscript $m$ represents one time step earlier than the present time denoted as superscript $m+1$; $\Delta x_w$, $\Delta x_e$, $\Delta y_n$ and $\Delta y_s$ are grid sizes beside a nodal point $(i, j, k)$ in the west, east, north and south, respectively; $\Delta z$ is the grid size in $z$-axis ; $\Delta t$ is the time step. Note that Eq. (43) reduces to the discretized expression of Eq. (6) when the quadratic terms are neglected. Similarly, Eq. (1) is approximated as

$$K_x \left( \frac{\Delta x_w}{\Delta X} h_{i+1,j,k}^{m+1} - \frac{2}{\Delta x_w \Delta x_e} h_{i,j,k}^{m+1} + \frac{\Delta x_e}{\Delta X} h_{i-1,j,k}^{m+1} \right) + K_y \left( \frac{\Delta y_s}{\Delta Y} h_{i,j+1,k}^{m+1} - \frac{2}{\Delta y_s \Delta y_n} h_{i,j,k}^{m+1} + \frac{\Delta y_n}{\Delta Y} h_{i,j-1,k}^{m+1} \right) + \frac{K_z}{\Delta z^2} \left( h_{i,j,k+1}^{m+1} - 2h_{i,j,k}^{m+1} + \right.$$

$$\left. h_{i,j,k-1}^{m+1} \right) = \frac{S_s}{\Delta t} \left( h_{i,j,k}^{m+1} - h_{i,j,k}^{m} \right) \tag{44}$$

where $h_{i,j,k}^{m}$ is the hydraulic head in the saturated zone at a nodal point $(i, j, k)$. The initial condition for each nodal point is expressed as

$$\phi_{i,j,k}^{1} = h_{i,j,k}^{1} = 0 \text{ at each } (i, j, k) \tag{45}$$

The no-flow condition specified at the outer boundaries shown in Fig. 2a and the bottom can be written as

$$\phi_{i-1,j,k}^{m+1} = \phi_{i+1,j,k}^{m+1} \text{ and } h_{i-1,j,k}^{m+1} = h_{i+1,j,k}^{m+1} \text{ at } i = 1, \, n_x \tag{46}$$

$$\phi_{i,j-1,k}^{m+1} = \phi_{i,j+1,k}^{m+1} \text{ and } h_{i,j-1,k}^{m+1} = h_{i,j+1,k}^{m+1} \text{ at } j = 1, \, n_y \tag{47}$$

$$h_{i,j,k-1}^{m+1} = h_{i,j,k+1}^{m+1} \text{ at } k = 1 \tag{48}$$

where $n_x$ and $n_y$ are the total number of grids in $x$- and $y$-axes, respectively. The top boundary condition is approximated as

$$\begin{cases} \phi_{i,j,k-1}^{m+1} = \phi_{i,j,k+1}^{m+1} & \text{outside the recharge area} \\ \frac{K_z e^{-ab}}{\Delta z} \left( \phi_{i,j,k+1}^{m+1} - \phi_{i,j,k}^{m+1} \right) = I & \text{inside the recharge area} \end{cases} \text{ at } k = n_z \tag{49}$$

where $n_z$ is the total number of grids in $z$-axis. The grid sizes $\Delta x_w$, $\Delta x_e$, $\Delta y_s$, and $\Delta y_n$ are all 5 m inside the recharge area while outside the area they gradually increase according to the formula: $\Delta x_e = 1.2 \Delta x_w$ and $\Delta y_n = 1.2 \Delta y_s$ starting from $\Delta x_w = \Delta y_s = 5$ m and $\Delta x_e = \Delta y_n = 1.2 \times 5$ m = 6 m. Note that the largest grid size is set equal to 25 m for good accuracy in solution prediction. The grid size $\Delta z$ is set to 0.1 m and the time step $\Delta t$ is chosen as 0.1 day for the period of 0~2.5 days and 0.25 day for 2.5~100 days. The total number of the nodal points is 327,789. The values of the hydraulic parameters are shown in Table 1.

The head solution to the nonlinear model of Eqs. (43) – (49) is obtained by an iteration method. Initially, the quadratic terms in Eq. (43) are assumed as $K_x g_x^{(n-1)} G_x^{(n)} + K_y g_y^{(n-1)} G_y^{(n)} + K_z g_z^{(n-1)} G_z^{(n)}$ with

$$G_x^{(n)} = \Delta x_w \phi_{i+1,j,k}^{m+1,(n)} - (\Delta x_w + \Delta x_e) \phi_{i,j,k}^{m+1,(n)} + \Delta x_e \phi_{i-1,j,k}^{m+1,(n)} \tag{50a}$$

$$G_y^{(n)} = \Delta y_s \phi_{i,j+1,k}^{m+1,(n)} - (\Delta y_s + \Delta y_n) \phi_{i,j,k}^{m+1,(n)} + \Delta y_n \phi_{i,j-1,k}^{m+1,(n)} \tag{50b}$$

$$G_z^{(n)} = (\phi_{i,j,k+1}^{m+1,(n)} - \phi_{i,j,k}^{m+1,(n)})/\Delta z \tag{50c}$$

$$g_x^{(n-1)} = \Delta x_w \phi_{i+1,j,k}^{m+1,(n-1)} - (\Delta x_w + \Delta x_e) \phi_{i,j,k}^{m+1,(n-1)} + \Delta x_e \phi_{i-1,j,k}^{m+1,(n-1)} \tag{50d}$$

$$g_y^{(n-1)} = \Delta y_s \phi_{i,j+1,k}^{m+1,(n-1)} - (\Delta y_s + \Delta y_n) \phi_{i,j,k}^{m+1,(n-1)} + \Delta y_n \phi_{i,j-1,k}^{m+1,(n-1)} \tag{50e}$$

$$g_z^{(n-1)} = (\phi_{i,j,k+1}^{m+1,(n-1)} - \phi_{i,j,k}^{m+1,(n-1)})/\Delta z \tag{50f}$$

where superscript $(n)$ represents the $n$-th iteration and gradients $g_x^{(n-1)}$, $g_y^{(n-1)}$ and $g_z^{(n-1)}$ cause a linearized Eq. (43) because they are known head values from the previous iteration. At the first time step (i.e., $t = \Delta t$, $m = 2$), the first iteration solves a system of Eqs. (44) – (49) and the linearized Eq. (43) with $g_x^{(0)} = g_y^{(0)} = g_z^{(0)} = 1$ and obtains the numerical solution of $\phi_{i,j,k}^{2,(1)}$ at each nodal point. The second iteration obtains $\phi_{i,j,k}^{2,(2)}$ with updated values of $g_x^{(1)}$, $g_y^{(1)}$ and $g_z^{(1)}$ from the previous result of $\phi_{i,j,k}^{2,(1)}$. Repeat this iteration process for $n \geq 3$ until the convergence condition of $\left| \phi_{i,j,k}^{2,(n)} - \phi_{i,j,k}^{2,(n-1)} \right| < 10^{-4}$ at each nodal point in the unsaturated zone is satisfied. The last result of $\phi_{i,j,k}^{2,(n)}$ is therefore the head solution to the nonlinear model. Similarly, the iteration process is applied to obtain $\phi_{i,j,k}^{m,(n)}$ for $m \geq 3$ at the other time steps (i.e., $t = 2\Delta t$, $3\Delta t$,...) with the convergence condition $\left| \phi_{i,j,k}^{m,(n)} - \phi_{i,j,k}^{m,(n-1)} \right| < 10^{-4}$. Note that the first iteration at each time step calculates $g_x^{(0)}$, $g_y^{(0)}$ and $g_z^{(0)}$ using $\phi_{i,j,k}^{m,(n)}$ obtained at the previous time step.

## 3 Results and discussion

The default values of the parameters and variables used in the calculation of the present solution are listed in Table 1. In Sect. 3.1, the error arising from neglecting the process of infiltration in the unsaturated zone is examined. In Sect. 3.2, the recharge efficiency associated with the properties of the unsaturated zone is investigated. In Sect. 3.3, the sensitivity analysis of the hydraulic head in the unsaturated zone in regard to various hydraulic parameters is discussed. In Sect. 3.4, the present solution is compared with a finite difference solution. In Sect. 3.5, the present solution is applied to a field problem of artificial recharge.

### 3.1 Effect of unsaturated flow on head distributions in aquifers

Here we investigate the difference between the present solution and Chang et al.'s (2016) analytical solution to explore the effect of unsaturated flow on the head distributions in the aquifer. Chang et al.'s (2016) solution considers 3D saturated flow in an unconfined aquifer with localized recharge but neglects the effect of unsaturated flow. One might expect that the difference is mainly dominated by the magnitudes of parameters $\alpha$ (dimensionless unsaturated exponent) and $\bar{b}$ (dimensionless unsaturated thickness). Figure 3 displays the predicted temporal head distributions at $(\bar{x}, \bar{y}, \bar{z}) = (2, 0, -0.5)$ by their solution and the present solution, Eq. (36b), for different pairs of $(\alpha, \bar{b})$ with $\alpha = 10^2$ or $10^3$ and $\bar{b}$ from $10^{-3}$ to one. Significant difference in $\bar{h}$ predicted by both solutions can be seen except the cases that $(\alpha, \bar{b}) = (10^2, 10^{-1})$, $(10^3, 10^{-1})$, and $(10^3, 10^{-2})$ shown in the figure. The result indicates that the thickness of the unsaturated zone is less than 10 % of the saturated aquifer thickness (i.e., $\bar{b} \leq 0.1$) for obtaining close predictions from both solutions. When $\bar{b} = 10^{-2}$, both solutions disagree if $\alpha = 10^2$ and agree well if $\alpha = 10^3$, indicating that the magnitude of the product $\alpha\bar{b} (= ab)$ should at least be 10 (i.e., $\alpha\bar{b} \geq 10$) for good agreement of both solutions. It is worth noting that both solutions disagree even for a much thinner

unsaturated zone as compared with the aquifer (i.e., $\bar{b} = 10^{-3}$) because of $\alpha\bar{b} < 10$. Additionally, the curves for $\alpha = 10$ all disaccord with Chang et al.'s (2016) solution, whereas the curves for $\alpha = 10^4$ match with this solution except two cases that $(\alpha, \bar{b}) = (10^4, 0.5)$ and $(10^4, 1)$ (not shown in the figure). Judging from the above, one can recognize the effect of unsaturated flow on the predicted head in saturated aquifers is negligible when $\alpha\bar{b} \geq 10$ and $\bar{b} \leq 0.1$. A great number of existing analytical solutions ignoring unsaturated flow give accurate predictions only when those two inequalities are satisfied (e.g., Chang and Yeh, 2007; Illas et al., 2008; Bansal and Teloglou, 2013). Otherwise, significant deviations will happen in their predictions.

### 3.2 Effect of unsaturated flow on recharge efficiency

The effect of the unsaturated zone on the *RE* is explored based on the curves of the *RE* versus $\bar{t}$ shown in Fig. 4 plotted using Eq. (40) for different pairs of $(\alpha, \bar{b}) = (100, 0.5)$, $(10, 0.5)$, $(1, 0.01)$, $(1, 0.5)$, and $(1, 1)$. For a given $\bar{t}$, the *RE* increases with $\alpha$ for a fixed $\bar{b}$ and decreasing $\bar{b}$ for a fixed $\alpha$. After $\bar{t} = 10^6$, the *RE* approaches an ultimate value equalling unity when $(\alpha, \bar{b}) = (100, 0.5)$ and $(1, 0.01)$, 0.9 when $(\alpha, \bar{b}) = (10, 0.5)$, 0.7 when $(\alpha, \bar{b}) = (1, 0.5)$, and 0.6 when $(\alpha, \bar{b}) = (1, 1)$. Those results imply that the ultimate recharge efficiency (*URE*) depends on the magnitudes of both $\alpha$ and $\bar{b}$. Figure 5 illustrates contours of the *URE* at $\bar{t} = 10^7$ in the ranges of $0.01 \leq \bar{b} \leq 1$ and $1 \leq \alpha \leq 100$. The *URE* approaches unity when $\bar{b} < 0.05$ or $\alpha > 20$. In contrast, it is below 0.9 and related to a given pair $(\alpha, \bar{b})$ when $\bar{b} > 0.1$ and $\alpha < 10$. It is clearly seen that the *RE* is great for a large $\alpha$ and/or a small $\bar{b}$. Those results provide useful information in the estimation of the amount of water from the recharge entering the aquifer. Notice that the case of *URE* < 1 may be due to the problem that unsaturated flow is influenced by the water retention capacity and diffusivity in the horizontal direction.

### 3.3 Sensitivity analysis for flow in unsaturated zone

Chang et al. (2016) performed the sensitivity analysis to investigate the sensitivity of the hydraulic head in saturated aquifers to the change in each of the aquifer parameters. This section focuses on the sensitivity analysis of the head in the unsaturated zone. Consider the recharge area of $0 \leq x \leq 50$ m and $0 \leq y \leq 50$ m and the observation points A at (0, 0, 5 m) under the area and B at (100 m, 0, 5 m) beside the area. Other values of the parameters are given in Table 1. The temporal distribution curves of the normalized sensitivity coefficient $S_{i,t}$ predicted by Eq. (42) to each of the parameters $a$, $l$, $w$, $S_s$, $S_y$, $K_x$, $K_y$, and $K_z$ are exhibited in Fig. 6a for point A and Fig. 6b for point B. At a given time, a positive $S_{i,t}$ means that the change in the specific parameter causes the increase in the head. In contrast, a negative $S_{i,t}$ signifies that the change leads to the head decrease. The magnitude of the head remains unchanged when $S_{i,t} = 0$. Obviously, the parameters $l$, $w$, $S_y$, $K_x$ and $K_y$ are important factors affecting the predicted head observed at points A and B, revealing that those parameters should be included in the flow model. The head at point A is sensitive to the changes in $a$ and $K_z$ but that at point B is insensitive. The result implies that unsaturated flow prevails under the recharge area but does not away from the area. In addition, the coefficient $S_{i,t}$ to $S_s$ almost equals zero over the entire recharge period, indicating that the change in $S_s$ does not affect the predicted head in the unsaturated zone.

### 3.4 Validation of present solution

The finite difference solution to the unsaturated-saturated flow model based on the nonlinear and linearized versions of Richards' equation, Eqs. (5) and (6), has been developed and described in Sect. 2.6. It is used to validate the present solution. Figure 7 demonstrates temporal head distributions observed at (25 m, 0, 5 m) and (25 m, 0, -10 m) under the recharge area and at (94.65 m, 0, 5 m) and (94.65 m, 0, -10 m) beside the area. The figure displays good agreement on the predicted head distributions from both the solutions. It is noteworthy that numerous attempts had been made by scholars to examine the accuracy of the linearized version of Richards' equation (e.g., Kroszynski and Dagan, 1975; Mishra and Neuman, 2010; Liang et al., 2017b). They also revealed that the linearized equation causes insignificant deviation on model predictions. We therefore conclude that the present solution is correctly developed and fairly predicts the hydraulic head for the unsaturated-saturated flow induced by localized recharge.

### 3.5 Application of present solution to field experiment

Bianchi and Haskell (1968) executed a field experiment of artificial recharge from two ponds on an alluvial fan in Fresno County, California. Pond No. 2 was within a square of 90 m $\times$ 90 m on the ground. The average recharge rate of the pond was 0.107 m d$^{-1}$. The initial water table was 6.4 m below the ground and 24.384 m over the impervious aquifer bottom. The entire recharge period was 10.92 days on record. The values of the aquifer parameters obtained from well test data were $K_x = 7.925$ m d$^{-1}$ and $S_y = 0.022$. There are 19 observation data of the water table rise beneath the center of the pond versus time shown in Fig. 8. We apply the least square method using the Mathematica built-in function FindRoot (Wolfram, 1996) to estimate five parameters $a$, $K_x = K_y$, $K_z$, $S_s$, and $S_y$ based on the data and present solution. The estimated values are $a = 0.388$ m$^{-1}$, $K_x = 5.642$ m d$^{-1}$, $K_z = 1.573$ m d$^{-1}$, $S_s = 5 \times 10^{-5}$ m$^{-1}$, and $S_y = 0.102$ which are all in the reasonable ranges of their parameter values; they are $0.2 \leq a \leq 5$ m$^{-1}$ (Philip, 1969), $8.64 \times 10^{-2} \leq K_x \leq 864$ m d$^{-1}$, $0.1K_x \leq K_z \leq 0.33K_x$, $10^{-5} \leq S_s B \leq 10^{-3}$, and $0.01 \leq S_y \leq 0.3$ for sandy aquifers (Freeze and Cherry, 1979, p. 604). Figure 8 demonstrates 19 observed data of the water table rise, the predictions from Glover's (1960) solution with $K_x = 7.925$ m d$^{-1}$ and $S_y = 0.022$ provided in Bianchi and Haskell (1968), and the present solution with the five estimated parameters. The Glover solution was developed by assuming that the flow is radially outward from a circular recharge pond with an equivalent area to the square of 90 m $\times$ 90 m and the unsaturated flow above water table is neglected. The predictions from the present solution agree well with the observed data, but those from Glover's solution do not, indicating that the effect of unsaturated flow had better be considered because $\alpha \bar{b} = 2.48$ and $\bar{b} = 0.26$ in this case do not satisfy the condition of $\alpha \bar{b} \geq 10$ and $\bar{b} \leq 0.1$ concluded in Sect. 3.1. From those discussed above, the present solution has been shown to be applicable to a real-world problem for unsaturated-saturated flow due to a recharge pond.

## 4 Concluding remarks

This study develops a novel mathematical model depicting 3D unsaturated-saturated flow for the process that surface water recharge passes through an unsaturated zone and flows down to an unconfined aquifer. The Richards equation is considered to delineate unsaturated flow induced by infiltration due to recharge from the ground surface. The Gardner model is used to describe the unsaturated soil characteristics. The transient groundwater flow equation is then employed to describe the rise of the hydraulic head in the aquifer in response to the water flow from the unsaturated zone. Both equations are coupled by the continuity equations of the head and flux at the water table. The head solution to the model is derived by means of the Laplace transform and Fourier cosine transform. The recharge efficiency defined as the percentage of the water from the recharge down to the aquifer is clearly discussed. The sensitivity analysis is performed to investigate the head response to the change in each of the hydraulic parameters in the unsaturated zone. The present solution agrees well with the finite difference solution on predicting the time-varying head for the unsaturated-saturated flow model. In addition, the present solution is applied to study the observed data from a field experiment conducted by Bianchi and Haskell (1968). On the basis of the studies obtained from the present solution, the following conclusions can be drawn:

1.  The effect of unsaturated flow on the hydraulic head in the aquifer is ignorable when the product of the unsaturated exponent ($a$) and initial unsaturated thickness ($b$) is greater than 10 (i.e., $ab \geq 10$) and the unsaturated thickness is less than 10 % of the initial aquifer thickness ($B$) (i.e., $b/B \leq 0.1$). Otherwise, the effect should be considered to avoid large deviations in calculating the head in the aquifer. Existing models considering only saturated flow can predict accurate results only when these two inequalities are satisfied.

2.  The recharge efficiency initially equals zero, increases with time, and finally approaches a constant value (below or equal to unity) depending on the values of $\alpha$ ($= aB$) and $\bar{b}$ ($= b/B$).

3.  The ultimate recharge efficiency approaches unity when $\bar{b} < 0.05$ or $\alpha > 20$ but less than 90 % when $\bar{b} > 0.1$ and $\alpha < 10$. In other words, the surface source supplies more recharge water to the aquifer if the unsaturated zone has a large $\alpha$ and/or a small $\bar{b}$.

4.  The results of the sensitivity analysis indicate that the parameter $a$, $l$, or $w$ causes positive influence but $S_y$, $K_x$, $K_y$, or $K_z$ produces negative impact on the predicted head in the unsaturated zone. The head under the recharge area is sensitive to the changes in $a$ and $K_z$ but that beside the area is not. Moreover, the head is rather insensitive to the change in $S_s$.

**Acknowledgements**

This research has been supported in part by the grants from the Fundamental Research Funds for the Central Universities (2018B00114) and the Taiwan Ministry of Science and Technology under the contract numbers MOST 105-2221-E-009-043-MY2 and MOST 106-2221-E-009-066. The authors are indebted to the thoughtful and helpful comments of the editor and two reviewers.

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

**Table 1.** Default values of variables and hydraulic parameters.

| Notation | Default value (unit) | Definition |
|---|---|---|
| $a$ | 0.5 m$^{-1}$ | Unsaturated exponent in Gardner's model for soil characteristics |
| $(b, B)$ | (10 m, 20 m) | Initial thicknesses of unsaturated and saturated zones, respectively |
| $C$ | None | Specific moisture capacity |
| $d$ | 50 m | Shortest distance between the edge of recharge area and observation point |
| $h$ | None | Hydraulic head in saturated zone |
| $I$ | 0.1 m d$^{-1}$ | Recharge rate |
| $k_r$ | None | Relative hydraulic conductivity |
| $(K_x, K_y, K_z)$ | (10 m d$^{-1}$, 10 m d$^{-1}$, 1 m d$^{-1}$) | Saturated hydraulic conductivity in $x$, $y$, and $z$ directions, respectively |
| $(l, w)$ | 50 m | Half of width of recharge area in $x$ and $y$ directions, respectively |
| $(S_s, S_y)$ | ($10^{-5}$ m$^{-1}$, 0.2) | Specific storage and specific yield, respectively |
| $t$ | None | Time |
| $(x, y, z)$ | None | Cartesian coordinates |
| $\phi$ | None | Hydraulic head in unsaturated zone |
| $(\bar{b}, \bar{l}, \bar{w})$ | (0.5, 1, 1) | ($b/B$, $l/d$, $w/d$) |
| $(\bar{h}, \bar{\phi}, \bar{t})$ | None | ($h/B$, $\phi/B$, $K_x t/(S_s d^2)$) |
| $(\bar{x}, \bar{y}, \bar{z})$ | None | ($x/d$, $y/d$, $z/B$) |
| $(\alpha, \kappa_y, \kappa_z)$ | (10, 1, 0.625) | ($aB$, $K_y/K_x$, $K_z d^2/(K_x B^2)$) |
| $(\xi, \sigma)$ | (0.1, 1000) | ($I/K_z$, $S_y/(S_s B)$) |

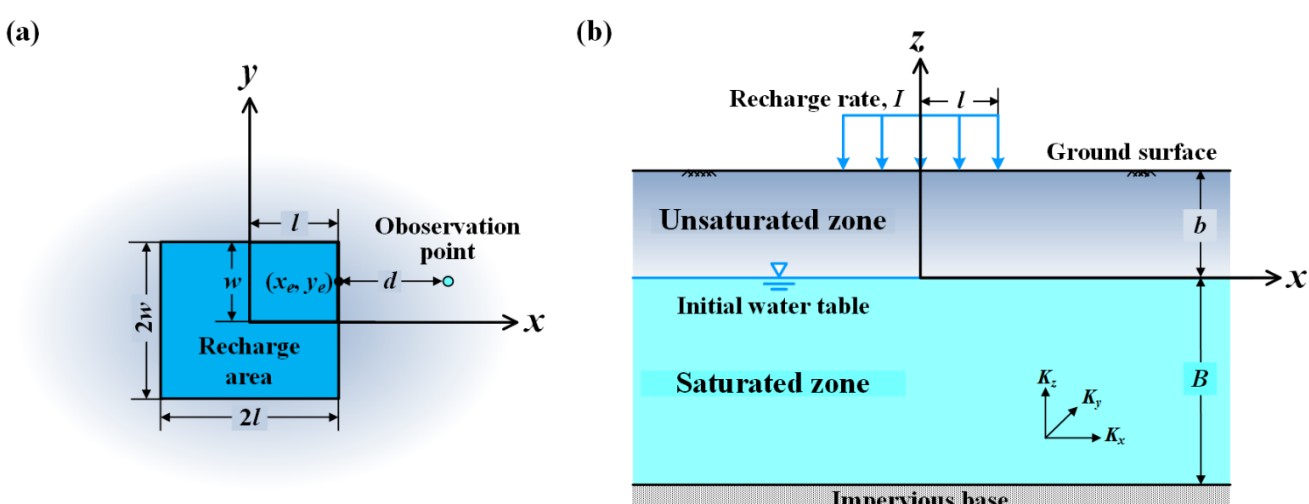

**Figure 1: Schematic diagram of unsaturated-saturated flow in an unconfined aquifer system with localized recharge (a) top view and (b) cross-sectional view.**

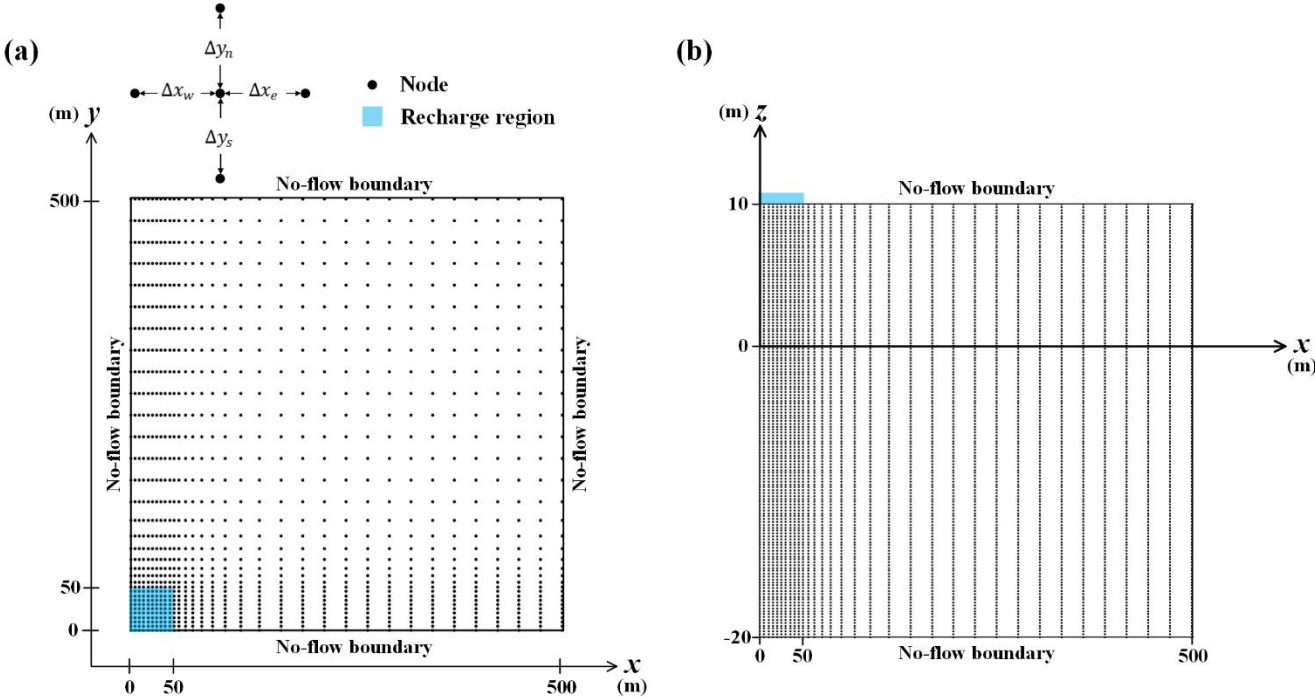

**Figure 2: Schematic diagram of finite difference grids (a) top view and (b) cross-sectional view.**

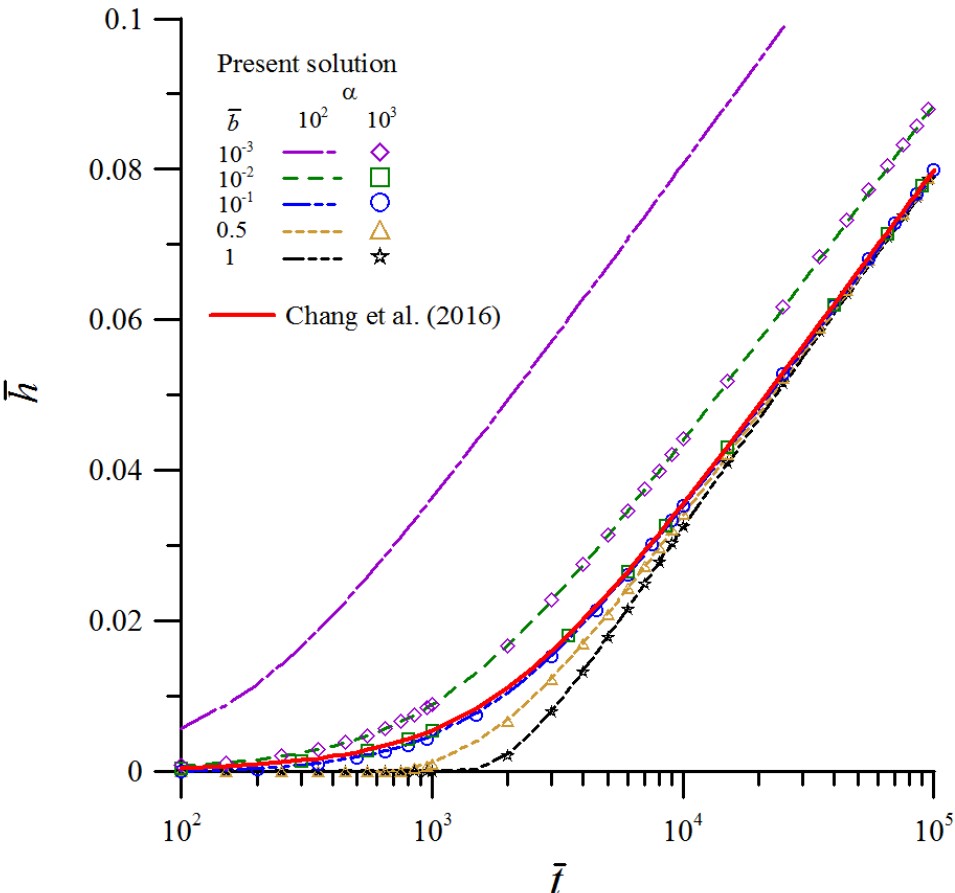

**Figure 3: Temporal distributios of the dimensionless head in the saturated zone predicted by the present solution and Chang et al.'s (2016) solution for different pairs of ($\alpha$, $\bar{b}$) representing the effect of unsaturated flow.**

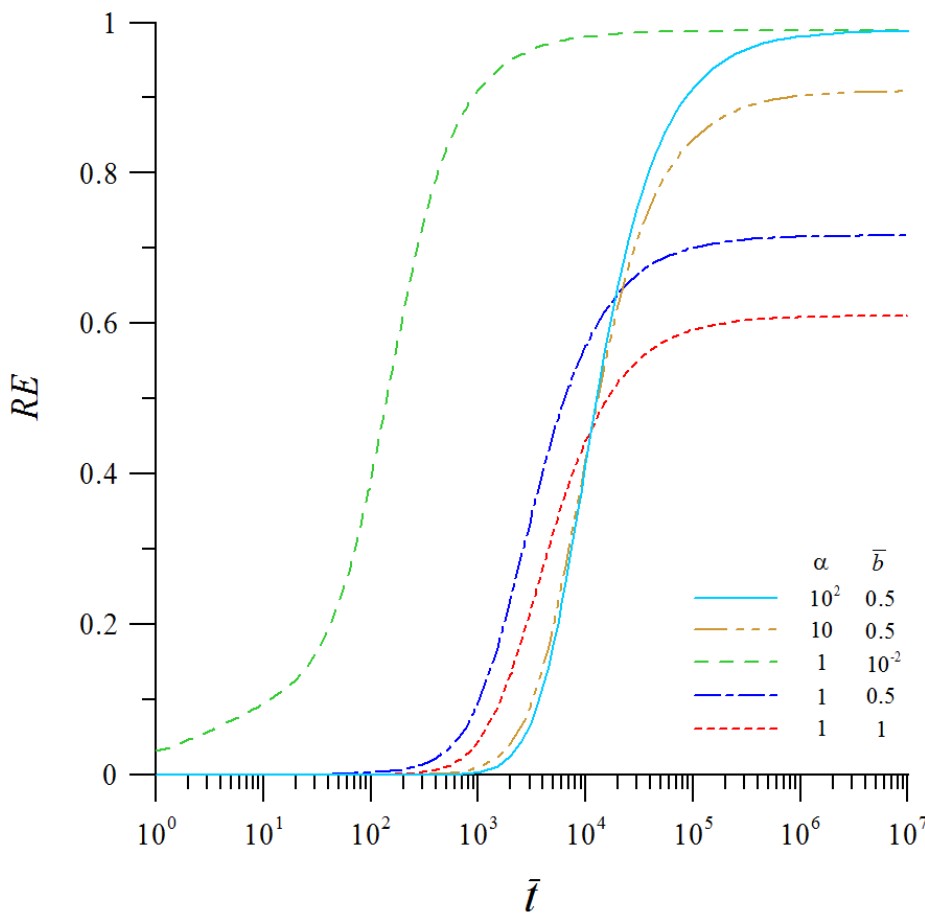

**Figure 4: Temporal distribution curves of the recharge efficiency (*RE*) for different pairs of ($\alpha$, $\bar{b}$) representing the effect of unsaturated flow.**

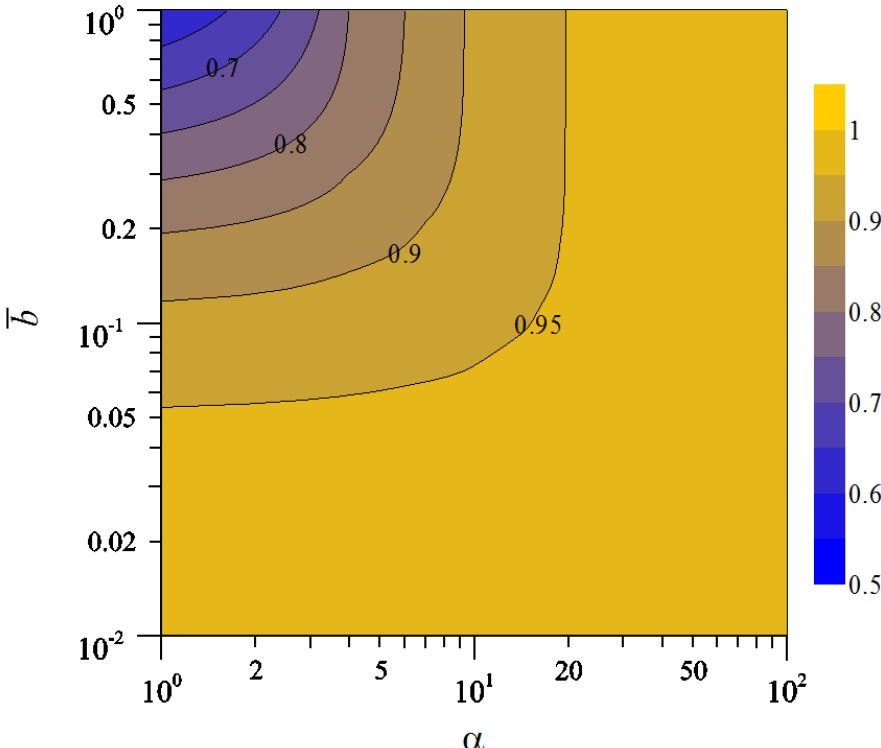

**Figure 5: Contours of the ultimate recharge efficiency (*URE*) plotted at $\bar{t} = 10^7$ for various values of $\alpha$ and $\bar{b}$.**

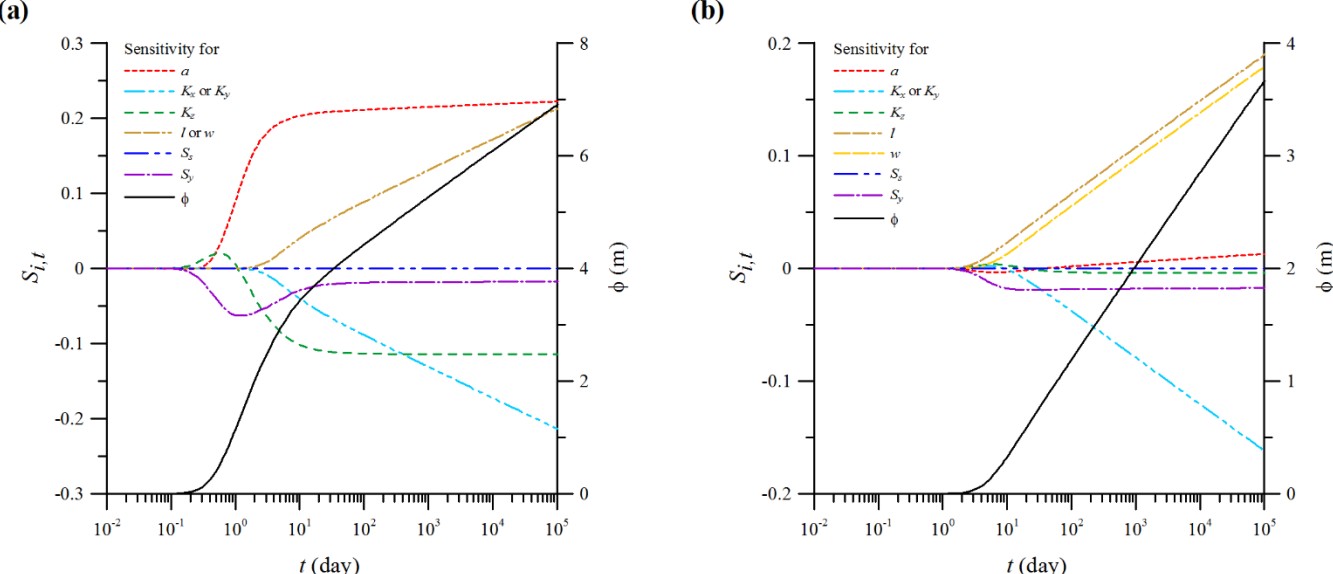

**Figure 6: Temporal distribution curves of the normalized sensitivity coefficients for the hydraulic head in unsaturated zone in response to the change in each of parameters *a*, *l*, *w*, $S_s$, $S_y$, $K_x$, $K_y$, and $K_z$ observed at (a) (0, 0, 5 m) under the recharge area and (b) (100 m, 0, 5 m) beside the area.**

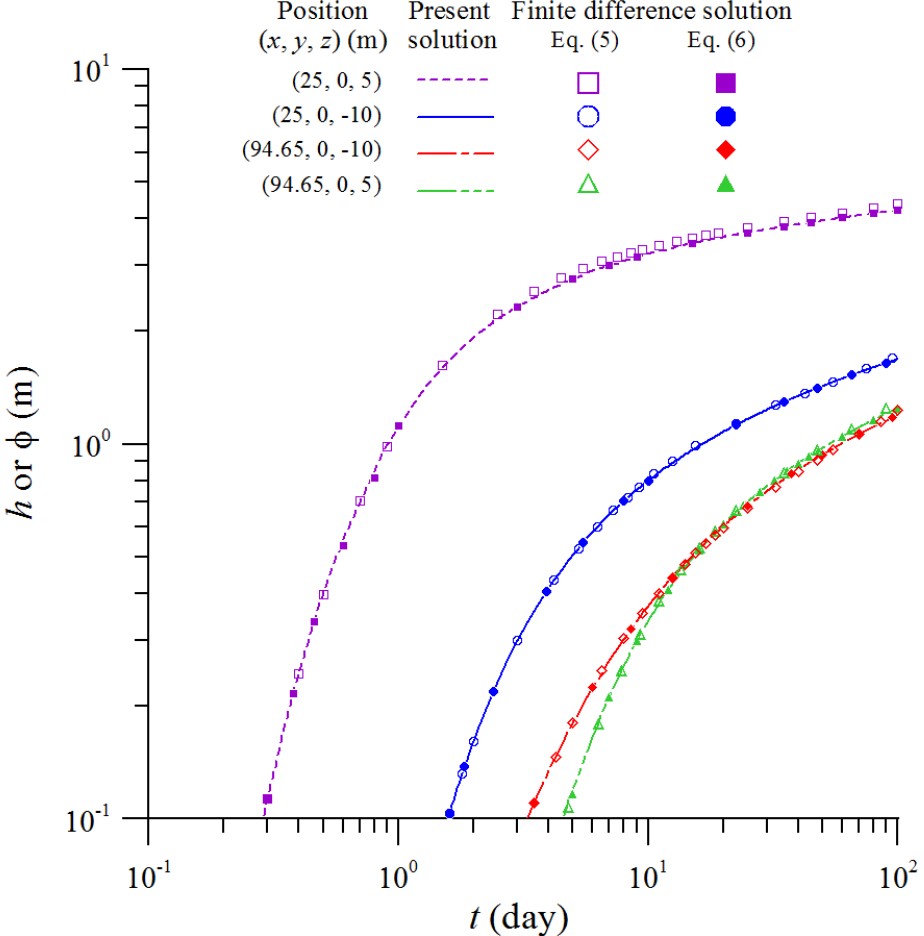

**Figure 7: Temporal distributions of the hydraulic head predicted by the present solution and the finite difference solution based on Richards' equation, Eq. (5), and its linearized version, Eq. (6), observed at (25 m, 0, 5 m) and (25 m, 0, -10 m) under the recharge area and at (94.65 m, 0, 5 m) and (94.65 m, 0, -10 m) beside the area.**

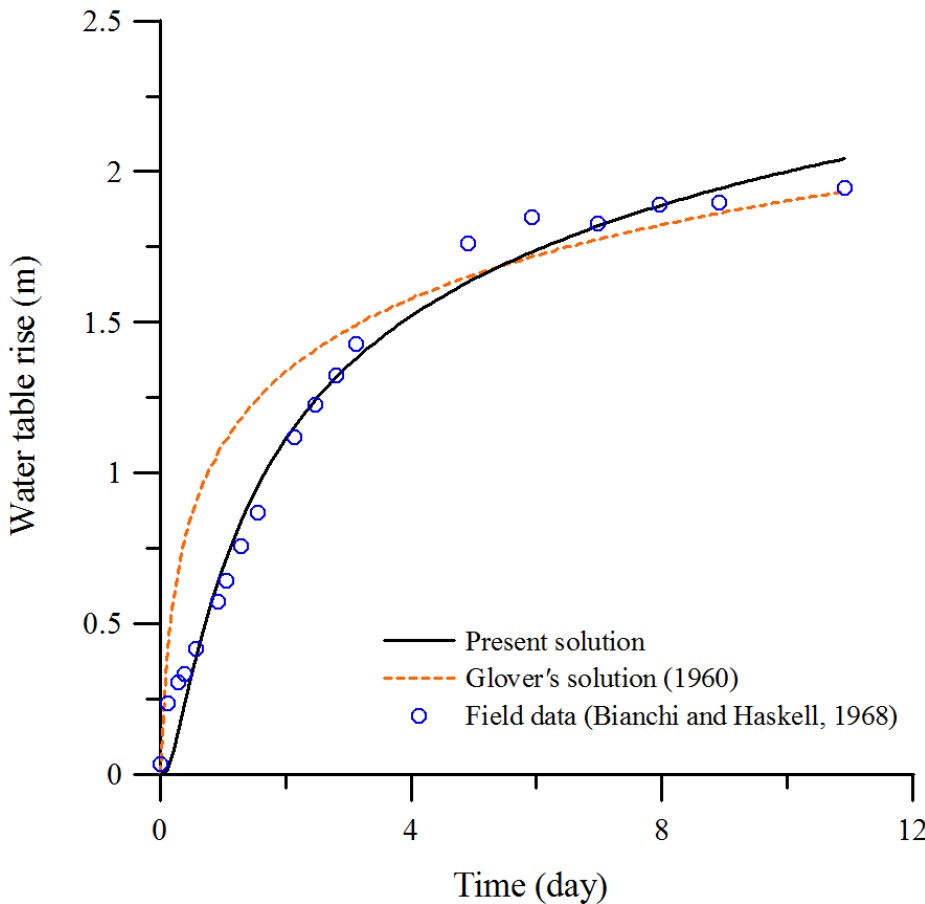

**Figure 8: Comparison of the water table rise predicted by the present solution and Glover's (1960) solution with field observed data given in Bianchi and Haskell (1968).**