# Peer review of "Analysis of three-dimensional unsaturated-saturated flow induced by localized recharge in unconfined aquifers"

_Hydrology and Earth System Sciences, 2017_

## Referee Comment (RC1) · Anonymous Referee #1 · 28 Nov 2017

The paper presents a three-dimensional semi-analytical solution that simulates flow in an unconfined aquifer as well as groundwater recharge.

Analytical solutions for solving the Richards equation are limited in the literature because, as these authors underline, the analytical solutions don't exist in most cases. However, this semi-analytical solution is identical to the one proposed by Tartakovsky and Neuman (2007), Mishra and Neuman (2010), Mishra and Neuman (2011), the authors could have cited their work in the introduction lines 25-26 (page 2), although, Mishra's solution was applied to pumping test. Authors should highlight the differences between their analytical solution and those proposed by Mishra et al., if there is any differences.

It is certainly common in hydrogeological modeling to consider recharge as an input to hydrodynamics models of aquifers. However, several studies have been dedicated to calculate recharge in the literature, these models are both empirical or conceptual (Sophocleous et Perkins 2000; Facchi et al. 2004; Markstrom et al. 2008) and physical solving the Richards equation ( Twarakavi et al. 2008; Thoms et al. 2006; Shen et Phanikumar 2010; Kuznetsov et al. 2012; Zhu et al. 2012). On contrary to what the authors stated in lines 9-10 (page 1), to be clear, they may have to add in the case of an analytical solution. Also, it is well known that the consideration of the unsaturated zone in the modeling of the recharge is important, unlike pumping. The recharge reflects the amount of water that comes from the precipitation and reaches the water table, this amount of water flow through the entire unsaturated zone. While in the case of pumping, water is directly extract from the saturated zone and several models neglect the contribution of the unsaturated zone located above. Hence the effects of the unsaturated zone in the case of pumping are discussed in the literature. I don't believe that studying the effect of the unsaturated zone in case of recharge will be something new, neither their analytical solution, since this one was already applied to pumping test. Also, the effects of Gardner parameters on the unsaturated zone flow have been discussed in (Mishra and Neuman 2011).

The paper is not well written, the English must be significantly improved. Mathematical equations aren't well written and test cases (and results) are not well described. Moreover, the first three conclusions drawn are not original. I don't recommend publication of this article.

Facchi A, Ortuani B, Maggi D, Gandolfi C (2004) Coupled SVAT–groundwater model for water resources simulation in irrigated alluvial plains. Environ Model Softw 19:1053–1063. doi: 10.1016/j.envsoft.2003.11.008

Kuznetsov M, Yakirevich A, Pachepsky YA, et al (2012) Quasi 3D modeling of water flow in vadose zone and groundwater. J Hydrol 450–451:140–149. doi: 10.1016/j.jhydrol.2012.05.025

Markstrom SL, Niswonger RG, Regan RS, et al (2008) GSFLOW, Coupled Ground-Water and Surface-Water Flow Model Based on the Integration of the Precipitation-Runoff Modeling System (PRMS) and the Modular Ground-Water Flow Model (MODFLOW-2005).

Mishra, P. K., and S. P. Neuman (2010), Improved forward and inverse analyses of saturated-unsaturated flow toward a well in a compressible unconfined aquifer, Water Resour. Res., 46(7), W07508, doi:10.1029/2009WR008899.

Mishra, P. K., and S. P. Neuman (2011), Saturated-unsaturated flow to a well with storage in a compressible unconfined aquifer, Water Resour. Res., 47(5), W05553, doi:10.1029/2010WR010177.

Shen C, Phanikumar MS (2010) A process-based, distributed hydrologic model based on a large-scale method for surface–subsurface coupling. Adv Water Resour 33:1524–1541. doi: 10.1016/j.advwatres.2010.09.002

Sophocleous M, Perkins SP (2000) Methodology and application of combined watershed and ground-water models in Kansas. J Hydrol 236:185–201. doi: 10.1016/S0022-1694(00)00293-6

Tartakovsky, G. D., and S. P. Neuman (2007), Three-dimensional saturated-unsaturated flow with axial symmetry to a partially penetrating well in a compressible unconfined aquifer, Water Resour. Res., 43(1), W01410, doi:10.1029/2006WR005153.

Thoms RB, Johnson RL, Healy RW, Geological Survey (U. S.) (2006) User's guide to the Variably Saturated Flow (VSF) process for MODFLOW [electronic resource] / by R. Brad Thoms, Richard L. Johnson, and Richard W. Healy. U.S. Dept. of the Interior, U.S. Geological Survey, Reston, Va

Twarakavi NKC, Šimůnek J, Seo S (2008) Evaluating Interactions between Groundwater and Vadose Zone Using the HYDRUS-Based Flow Package for MODFLOW. Vadose Zone J 7:757. doi: 10.2136/vzj2007.0082

Zhu Y, Shi L, Lin L, et al (2012) A fully coupled numerical modeling for regional unsaturated–saturated water flow. J Hydrol 475:188–203. doi: 10.1016/j.jhydrol.2012.09.048

---

## Author Comment (AC1) · 18 Dec 2017

**Responses to Comment of Referee #1**

The paper presents a three-dimensional semi-analytical solution that simulates flow in an unconfined aquifer as well as groundwater recharge.

Analytical solutions for solving the Richards equation are limited in the literature because, as these authors underline, the analytical solutions don't exist in most cases. However, this semi-analytical solution is identical to the one proposed by Tartakovsky and Neuman (2007), Mishra and Neuman (2010), Mishra and Neuman (2011), the authors could have cited their work in the introduction lines 25-26 (page 2), although, Mishra's solution was applied to pumping test. Authors should highlight the differences between their analytical solution and those proposed by Mishra et al., if there is any differences.

Response: Thanks for the comment. The difference between our solution and theirs is that we consider groundwater recharge problems for a localized recharge from the ground surface as a plane source to the aquifer while they study pumping drawdown problems for an extraction well treated as a line sink in the aquifer. In addition, the governing equations (GEs) describing flow in both saturated and unsaturated zones in our study are three-dimensional expressed in Cartesian coordinates while their flow equations for saturated and unsaturated zones are two-dimensional written in cylindrical coordinates. In addition, our solution was derived by applying the Fourier cosine transform to the GEs, but theirs was developed based on the Hankel transform. Therefore, both solutions are completely different in mathematical forms. We added the following sentences in the revised manuscript: "Such a coupled flow model has been applied to study drawdown behaviors induced by well pumping (e.g., Tartakovsky and Neuman, 2007; Mishra and Neuman, 2010; Mishra and Neuman, 2011). This paper investigates spatiotemporal distribution of the hydraulic head affected by localized recharge from the ground surface."

It is certainly common in hydrogeological modeling to consider recharge as an input to hydrodynamics models of aquifers. However, several studies have been dedicated to calculate recharge in the literature, these models are both empirical or conceptual (Sophocleous et Perkins 2000; Facchi et al. 2004; Markstrom et al. 2008) and physical solving the Richards equation ( Twarakavi et al. 2008; Thoms et al. 2006; Shen et Phanikumar 2010; Kuznetsov et al. 2012; Zhu et al. 2012). On contrary to what the authors stated in lines 9-10 (page 1), to be clear, they may have to add in the case of an analytical solution. Also, it is well known that the consideration of the unsaturated zone in the modeling of the recharge is important, unlike pumping. The recharge reflects the amount of water that comes from the precipitation and reaches the water table, this amount of water flow through the entire unsaturated zone. While in the case of pumping, water is directly extract from the saturated zone and several models neglect the contribution of the unsaturated zone located above. Hence the effects of the unsaturated zone in the case of pumping are discussed in the literature. I don't believe that studying the effect of the unsaturated zone in case of recharge will be something new, neither their analytical solution, since this one was already applied to pumping test. Also, the effects of Gardner parameters on the unsaturated zone flow have been discussed in (Mishra and

Neuman 2011).

Response: The sentence in the abstract of the original manuscript (lines 9-10, page 1) "Up to now, little attention has been given to the effect of unsaturated flow on the hydraulic head within the aquifer due to recharge." is changed as: "Little attention has been given to the development of analytical solutions to a coupled saturated-unsaturated flow due to a localized recharge up to now."

Although many studies have investigated unsaturated flow for groundwater recharge, our work has the following three novelties:

1. Our semi-analytical solution is indeed a new one that has not been seen in the literature. We develop the solution because it can serve as a preliminary design tool for the development of water resources management or groundwater remediation plan or a primary mean for testing and benchmarking numerical codes.

2. Analyses of quantitative results are presented based on the present solution and absolutely not found in the literature. The results demonstrate that the present solution is capable of exploring the insight into how the unsaturated flow affects the recharge efficiency and head distributions in the saturated zone. Please refer to the next response for details.

3. Sensitivity analysis assesses the response of the hydraulic head in the unsaturated zone to the change in each of the hydraulic parameters, especially the parameter associated with unsaturated flow. Please refer to the fourth conclusion in section 4 of the original manuscript.

The paper is not well written, the English must be significantly improved. Mathematical equations aren't well written and test cases (and results) are not well described. Moreover, the first three conclusions drawn are not original. I don't recommend publication of this article.

Response: This manuscript will be edited by a colleague who is good at English writing. The first three conclusions given below illustrate quantitative results which are new findings and absolutely not seen elsewhere. The first conclusion quantifies the validity of neglecting the effect of unsaturated flow on the hydraulic head in the underlying aquifer. Existing analytical solutions neglecting unsaturated flow give accurate predictions only when the quantitative conditions (i.e., $ab \geq 10$ and $b/B \leq 0.1$) are satisfied (e.g., Chang and Yeh, 2007; Illas et al., 2008; Bansal and Teloglou, 2013). Otherwise, significant deviations may happen to their predictions. The second and third ones propose a quantitative condition (i.e., $\bar{b} < 0.05$ or $\alpha > 20$) that causes almost all amount of localized recharge reaches the aquifer.

1. The effect of unsaturated flow on the hydraulic head in the aquifer is ignorable when the product of the unsaturated exponent ($a$) and initial unsaturated thickness ($b$) is greater than 10 (i.e., $ab \geq 10$) and the unsaturated thickness is less than 10 % of the initial aquifer thickness ($B$) (i.e., $b/B \leq 0.1$). Otherwise, the effect should be considered to avoid large deviations in calculating the head in the aquifer. Existing models considering only saturated flow can predict accurate results only when these two inequalities are

satisfied.

2. The recharge efficiency initially equals zero, increases with time, and finally approaches a constant value (below or equal to unity) depending on the values of $\alpha$ ($= aB$) and $\bar{b}$ ($= b/B$).

3. The ultimate recharge efficiency approaches unity when $\bar{b} < 0.05$ or $\alpha > 20$ but less than 90 % when $\bar{b} > 0.1$ and $\alpha < 10$. In other words, the surface source supplies more recharge water to the aquifer if the unsaturated zone has a large $\alpha$ and/or a small $\bar{b}$.

**References**

Bansal, R. K., and Teloglou, I. S.: An Analytical Study of Groundwater Fluctuations in Unconfined Leaky Aquifers Induced by Multiple Localized Recharge and Withdrawal, Global Nest J, 15, 394-407, 2013.

Chang, Y. C., and Yeh, H. D.: Analytical solution for groundwater flow in an anisotropic sloping aquifer with arbitrarily located multiwells, J Hydrol, 347, 143-152, 2007.

Facchi, A., Ortuani, B., Maggi, D., and Gandolfi, C.: Coupled SVAT-groundwater model for water resources simulation in irrigated alluvial plains, Environ Modell Softw, 19, 1053-1063, 10.1016/j.envsoft.2003.11.008, 2004.

Illas, T. S., Thomas, Z. S., and Andreas, P. C.: Water table fluctuation in aquifers overlying a semi-impervious layer due to transient recharge from a circular basin, J Hydrol, 348, 215-223, 2008.

Kuznetsov, M., Yakirevich, A., Pachepsky, Y. A., Sorek, S., and Weisbrod, N.: Quasi 3D modeling of water flow in vadose zone and groundwater, J Hydrol, 450, 140-149, 10.1016/j.jhydrol.2012.05.025, 2012.

Markstrom, S.L., Niswonger, R.G., Regan, R.S., Prudic, D.E., and Barlow, P.M.: GSFLOW, Coupled Ground-Water and Surface-Water Flow model based on the integration of the Precipitation-Runoff Modeling System (PRMS) and the Modular Ground-Water Flow Model (MODFLOW-2005) U.S. Geological Survey Techniques and Methods 6-D1, 240 p., 2008.

Mishra, P. K., and Neuman, S. P.: Improved forward and inverse analyses of saturated-unsaturated flow toward a well in a compressible unconfined aquifer, Water Resour Res, 46, Artn W07508 10.1029/2009wr008899, 2010.

Mishra, P. K., and Neuman, S. P.: Saturated-unsaturated flow to a well with storage in a compressible unconfined aquifer, Water Resour Res, 47, Artn W05553 10.1029/2010wr010177, 2011.

Shen, C. P., and Phanikumar, M. S.: A process-based, distributed hydrologic model based on a large-scale method for surface-subsurface coupling, Adv Water Resour, 33, 1524-1541, 10.1016/j.advwatres.2010.09.002, 2010.

Sophocleous, M., and Perkins, S. P.: Methodology and application of combined watershed and ground-water models in Kansas, J Hydrol, 236, 185-201, Doi 10.1016/S0022-1694(00)00293-6, 2000.

Tartakovsky, G. D., and Neuman, S. P.: Three-dimensional saturated-unsaturated flow with axial symmetry to a partially penetrating well in a compressible unconfined aquifer, Water Resour Res, 43, Artn

W01410 10.1029/2006wr005153, 2007.

Thoms, R.B., Johnson, R.L., and Healy, R.W.: User's guide to the variably saturated flow (VSF) Process for MODFLOW: U.S. Geological Survey Techniques and Methods 6-A18, 58 p., 2006.

Twarakavi, N. K. C., Simunek, J., and Seo, S.: Evaluating interactions between groundwater and vadose zone using the HYDRUS-based flow package for MODFLOW, Vadose Zone J, 7, 757-768, 10.2136/vzj2007.0082, 2008.

Zhu, Y., Shi, L. S., Lin, L., Yang, J. Z., and Ye, M.: A fully coupled numerical modeling for regional unsaturated-saturated water flow, J Hydrol, 475, 188-203, 10.1016/j.jhydrol.2012.09.048, 2012.

---

## Referee Comment (RC2) · Anonymous Referee #2 · 27 Dec 2017

The manuscript presents a 3D coupled (semi-)analytical model that simulates flow in both the unsaturated and saturated zones with localized recharge on the ground surface. The authors used a simplified model for flow through the unsaturated zone based on the linearized Richards´ equation. The resulting system of linear partial differential equation is then solved using the Laplace transform to eliminate the time derivative and the double Fourier transform which convert the original system of PDEs to a system of ordinary differential equation. They obtained the expressions of the heads in the unsaturated and saturated zones in terms of infinite double integrals. The final solution is presented in the Laplace domain and then numerical inversion of Laplace transform is required to get the solution in the time space.

The manuscript overall is well written and clear but it could be improved. The technique used is standard (Laplace transform coupled with cosine Fourier transform) for such kind of coupled problems, but I think the work addressed is very interesting and the topic, in my opinion, is appropriate for HESS.

The consideration of the unsaturated zone in the modeling of recharge is important and to the best of my knowledge this subject has never been addressed analytically. Although coupled unsaturated/saturated flow model have been already addressed analytically in the framework of pumping tests (see for instance, Mathias and Bulter (2006), Mishra and Neuman (2010, 2011), Tartakovsky and Neuman (2007)), the mathematical model presented here is different (3D cartesian). In think this work can be considered as new contribution.

I have however two comments that merits to be mentioned and discussed in a revised version of the manuscript:

1. The authors should mention the above cited works and discuss how they differ from their present work.
2. In section 3.4, the authors compare the proposed analytical solution to a finite element numerical solution obtained from the NDsolve function of Mathematica. They used the linearized system of equations (1)-(11) in the numerical solver. This is good to show the correctness of the analytical solution. However, It would be interesting to perform a comparison between a numerical solution based on the original "nonlinear" Richards´ equation to investigate the effect of linearization on the head distribution in the unsaturated and saturated zones. This may affects also the general results of the

manuscripts. I think, it would not need too much efforts to include the nonlinear Richards´ equation in the NDsolve function of Mathematica.

The original nonlinear Richards´ equation writes as follows (Kroszynski and Dagan, 1975)

$$K_x \frac{\partial}{\partial x}\left(K(\phi)\frac{\partial \phi}{\partial x}\right) + K_y \frac{\partial}{\partial y}\left(K(\phi)\frac{\partial \phi}{\partial y}\right) + K_z \frac{\partial}{\partial z}\left(K(\phi)\frac{\partial \phi}{\partial z}\right) = C(\phi)\frac{\partial \phi}{\partial t}$$

with $K(\phi) = e^{\kappa(\phi-\phi_a)}$ and $C(\phi) = S_y \kappa e^{\kappa(\phi-\phi_a)}$

**References**

Kroszynski, U. I., and G. Dagan (1975), Well pumping in unconfined aquifers: The influence of the unsaturated zone, Water Resour. Res., 2(3), 479-490.

Mathias, S. A., and A. P. Butler (2006), Linearized Richards´equation approach to pumping test analysis in compressible aquifers, Water Resour. Res., 42(6), W06408, doi: 10.1029/2005WR004680.

Mishra, P. K., and S. P. Neuman (2010), Improved forward and inverse analyses of saturated-unsaturated flow toward a well in a compressible unconfined aquifer, Water Resour. Res., 46(7), W07508, doi:10.1029/2009WR008899.

Mishra, P. K., and S. P. Neuman (2011), Saturated-unsaturated flow to a well with storage in a compressible unconfined aquifer, Water Resour. Res., 47(5), W05553, doi:10.1029/2010WR010177.

Tartakovsky, G. D., and S. P. Neuman (2007), Three-dimensional saturated-unsaturated flow with axial symmetry to a partially penetrating well in a compressible unconfined aquifer, Water Resour. Res., 43(1), W01410, doi:10.1029/2006WR005153.

---

## Author Comment (AC2) · 29 Mar 2018

**Responses to Comment of Referee #2**

The manuscript presents a 3D coupled (semi-)analytical model that simulates flow in both the unsaturated and saturated zones with localized recharge on the ground surface. The authors used a simplified model for flow through the unsaturated zone based on the linearized Richards´ equation. The resulting system of linear partial differential equation is then solved using the Laplace transform to eliminate the time derivative and the double Fourier transform which convert the original system of PDEs to a system of ordinary differential equation. They obtained the expressions of the heads in the unsaturated and saturated zones in terms of infinite double integrals. The final solution is presented in the Laplace domain and then numerical inversion of Laplace transform is required to get the solution in the time space.

The manuscript overall is well written and clear but it could be improved. The technique used is standard (Laplace transform coupled with cosine Fourier transform) for such kind of coupled problems, but I think the work addressed is very interesting and the topic, in my opinion, is appropriate for HESS. The consideration of the unsaturated zone in the modeling of recharge is important and to the best of my knowledge this subject has never been addressed analytically. Although coupled unsaturated/saturated flow model have been already addressed analytically in the framework of pumping tests (see for instance, Mathias and Bulter (2006), Mishra and Neuman (2010, 2011), Tartakovsky and Neuman (2007)), the mathematical model presented here is different (3D cartesian). In think this work can be considered as new contribution.

Response: Thanks for the comment.

I have however two comments that merits to be mentioned and discussed in a revised version of the manuscript:

1. The authors should mention the above cited works and discuss how they differ from their present work.

Response: Thanks for the suggestion. We added the following sentences in the revised manuscript to address the differences between our work and theirs:

"Such a coupled flow model has been proposed to investigate pumping drawdown problems by several articles (e.g., Mathias and Bulter, 2006; Tartakovsky and Neuman, 2007; Mishra and Neuman, 2010; Mishra and Neuman, 2011). They treated an extraction well as a line sink in the aquifer while we consider the localized recharge as a plane source to the aquifer. The coupled flow model in their studies is 2D written in cylindrical coordinates while that in ours is 3D expressed in Cartesian coordinates. In addition, their solutions are obtained by the Hankel transform, but ours is based on the Fourier cosine

transform. The present work aims to investigate the spatiotemporal distribution of the hydraulic head due to localized recharge from the ground surface."

2. In section 3.4, the authors compare the proposed analytical solution to a finite element numerical solution obtained from the NDsolve function of Mathematica. They used the linearized system of equations (1)-(11) in the numerical solver. This is good to show the correctness of the analytical solution. However, It would be interesting to perform a comparison between a numerical solution based on the original "nonlinear" Richards´ equation to investigate the effect of linearization on the head distribution in the unsaturated and saturated zones. This may affects also the general results of the manuscripts. I think, it would not need too much efforts to include the nonlinear Richards´ equation in the NDsolve function of Mathematica.

The original nonlinear Richards´ equation writes as follows (Kroszynski and Dagan, 1975)

$$K_x \frac{\partial}{\partial x}\left(K(\phi)\frac{\partial \phi}{\partial x}\right) + K_y \frac{\partial}{\partial y}\left(K(\phi)\frac{\partial \phi}{\partial y}\right) + K_z \frac{\partial}{\partial z}\left(K(\phi)\frac{\partial \phi}{\partial z}\right) = C(\phi)\frac{\partial \phi}{\partial t}$$

with $K(\phi) = e^{\kappa(\phi-\phi_a)}$ and $C(\phi) = S_y\kappa e^{\kappa(\phi-\phi_a)}$

Response: Thanks for the suggestion. In recent days, we try very hard to solve the nonlinear Richards' equation (NRE) in our unsaturated-saturated coupled flow model using the NDSolve function of Mathematica. The associated computation consumes tremendous computing time in each computer run. The outputs, however, fail to yield reasonable results although we had tried a variety of measures to solve the NRE using fine grids, small time step, and different built-in options of the NDSolve function. A great deal of study has examined the accuracy of the linearized Richards' equation (LRE) (e.g., Kroszynski and Dagan, 1975; Tartakovsky and Neuman, 2007; Mishra and Neuman, 2010; Liang et al., 2017). Kroszynski and Dagan (1975) and Mishra and Neuman (2010), for example, achieved good agreement on aquifer drawdown estimated from an analytical solution based on the LRE and a numerical solution based on the NRE. Liang et al. (2017) also achieved agreement on the hydraulic head predicted by analytical and numerical solutions based on the LRE and the NRE, respectively. Tartakovsky and Neuman (2007) revealed that aquifer drawdown calculated by an analytical solution based on the LRE agrees well with that obtained in a field pumping test. We, therefore, add the following sentences in lines 2-5, page 11 of the revised manuscript:

"On the other hand, numerous attempts have been made by scholars to examine the accuracy and/or applicability of the linearized Richards' equation for Eqs. (2) − (4) (e.g., Kroszynski and Dagan, 1975; Tartakovsky and Neuman, 2007; Mishra and Neuman, 2010; Liang et al., 2017). Results show that the linearization of Richards' equation causes insignificant deviation on model prediction."

**References**

Kroszynski, U. I., and Dagan, G.: Well pumping in unconfined aquifers: The influence of the unsaturated zone, Water Resour Res, 11, 479-490, 10.1029/WR011i003p00479, 1975.

Liang, X. Y., Zhan, H. B., Zhang, Y. K., and Schilling, K.: Base flow recession from unsaturated-saturated porous media considering lateral unsaturated discharge and aquifer compressibility, Water Resour Res, 53, 7832-7852, 10.1002/2017WR020938, 2017.

Mathias, S. A., and Butler, A. P.: Linearized Richards' equation approach to pumping test analysis in compressible aquifers, Water Resour Res, 42, 2006.

Mishra, P. K., and Neuman, S. P.: Improved forward and inverse analyses of saturated-unsaturated flow toward a well in a compressible unconfined aquifer, Water Resour Res, 46, Artn W07508 10.1029/2009wr008899, 2010.

Mishra, P. K., and Neuman, S. P.: Saturated-unsaturated flow to a well with storage in a compressible unconfined aquifer, Water Resour Res, 47, Artn W05553 10.1029/2010wr010177, 2011.

Tartakovsky, G. D., and Neuman, S. P.: Three-dimensional saturated-unsaturated flow with axial symmetry to a partially penetrating well in a compressible unconfined aquifer, Water Resour Res, 43, Artn W01410 10.1029/2006wr005153, 2007.